# Reliance on self-reports and estimated food composition data in nutrition research introduces significant bias that can only be addressed with biomarkers

Javier I Ottaviani[1], Virag Sagi-Kiss[2], Hagen Schroeter[1], Gunter GC Kuhnle[3]*

[1]Mars, Incorporated, McLean, United States; [2]Imperial College London, London, United Kingdom; [3]University of Reading, Reading, United Kingdom

**Abstract** The chemical composition of foods is complex, variable, and dependent on many factors. This has a major impact on nutrition research as it foundationally affects our ability to adequately assess the actual intake of nutrients and other compounds. In spite of this, accurate data on nutrient intake are key for investigating the associations and causal relationships between intake, health, and disease risk at the service of developing evidence-based dietary guidance that enables improvements in population health. Here, we exemplify the importance of this challenge by investigating the impact of food content variability on nutrition research using three bioactives as model: flavan-3-ols, (–)-epicatechin, and nitrate. Our results show that common approaches aimed at addressing the high compositional variability of even the same foods impede the accurate assessment of nutrient intake generally. This suggests that the results of many nutrition studies using food composition data are potentially unreliable and carry greater limitations than commonly appreciated, consequently resulting in dietary recommendations with significant limitations and unreliable impact on public health. Thus, current challenges related to nutrient intake assessments need to be addressed and mitigated by the development of improved dietary assessment methods involving the use of nutritional biomarkers.

*For correspondence:
g.g.kuhnle@reading.ac.uk

## eLife assessment

This **important** study, using three bioactive compounds as a model, demonstrates that estimating the intake of food components based on food composition databases and self-reported dietary data is highly unreliable. The authors present **convincing** data showing the differences in the estimated quantile of intake of three bioactive compounds between biomarker and 24-hour dietary recall with food composition database. The work will be of broad interest to the clinical nutrition research community.

## Introduction

Nutrition is a crucial factor for public health (*Afshin et al., 2019*; *The National Academies of Sciences and Engineering and Medicine Health, 2017*). However, despite considerable methodological progress, nutrition research still relies mostly on self-reported dietary information and limited food composition data to investigate the links between health and nutrition. Indeed, food composition data is the bedrock on which nutrition research rests today: it allows us to estimate the intake of specific nutrients and other dietary compounds, and thus enables investigations into the associations between nutrient intake and health outcomes. Such data inform policymakers in the development of dietary

**eLife digest** Studies about the health benefits of foods or nutrients are often inconsistent. One study may find a health benefit of a particular food and may recommend that people increase their consumption of this food to reduce their disease risk. Yet another study may find the opposite. Inconsistent study results fuel confusion and frustration, and reduce trust in research.

Limitations in the studies' designs are likely to be blamed for the inconsistent findings. For example, many studies rely on participants to self-report their food intake and on databases of the nutritional content of food. But people may not accurately report their food intake. Foods vary in their nutritional content, even between two items of the same food such as two apples. And how individuals metabolize foods can further affect the nutrients they receive.

Nutritional biomarkers are a potential alternative to measuring dietary intake of specific nutrients. Biomarkers are compounds the body produces when it metabolizes a specific nutrient. Measuring biomarkers therefore give scientists a more accurate and unbiased assessment of nutrient intake.

Ottaviani et al. conducted a study to test the differences when estimating nutrient intake using nutritional biomarkers compared with more conventional tools. They analyzed data from a nutrition study that involved over 18,000 participants. The experiments used computer modelling to assess study results using self-reported food intake in combination with food composition database information, or measures of three biomarkers estimating the intake of flavan-3-ols, epicatechin, and nitrates. The models showed that self-reported intake and food database information often led to inaccurate results that did not align well with biomarker measurements.

Measuring nutritional biomarkers provides a more accurate and unbiased assessment of nutritional intake. Using these measurements instead of traditional methods for measuring nutrient intake may help increase the reliability of nutrition research. Scientists must work to identify and confirm biomarkers of nutrients to facilitate this work. Using these more precise nutrient measurements in studies may result in more consistent results. It may also lead to more trustworthy recommendations for consumers.

recommendations and risk assessments, and support the development of guidance for the general public and the food industry. However, this approach is not without significant challenges and limitations. One key challenge is the construction and maintenance of food composition data that underpin intake assessments for specific nutrients as foods are highly complex and widely variable in their chemical makeup. Multiple factors affect the ultimate nutrient content of foods, including cultivar or breed, climate, growing and harvest conditions, storage, processing, and methods of culinary preparation (*Greenfield and Southgate, 2003*). Even apples harvested at the same time from the self-same tree show more than a twofold difference in the amount of many micronutrients (*Wilkinson and Perring, 1961*). Moreover, processed foods are usually not standardised for composition but taste, texture, and consumer preferences, and thus vary in their chemical composition. Significant efforts have been made to generate extensive and detailed food composition tables, and complex sampling paradigms are used to obtain representative samples. Despite all these efforts, food composition data are generally used by relying on single-point estimates, the mean food composition, de facto assuming that foods have a consistent composition. This approach introduces a considerable degree of error, bias, and uncertainty – and these are exacerbated by the limitations of self-reported dietary data which are known to carry substantial bias (*Subar et al., 2015*).

Moreover, current approaches also assume that intake directly correlates with the systemic presence of a given nutrient as it is through their systemic presence that many nutrients mediate much of their health-related biological effects. This introduces even more complexity when assessing true nutrient intake as inter- and intra-individual aspects of absorption, metabolism, distribution, and excretion, processes also impacted by the gut microbiome and other potentially highly variable and individual modulators of nutrient levels in the human body, should ideally be taken into account.

While all of this is well known in the nutrition expert community (*Gibney et al., 2020*), the impact on both the interpretation of research findings and the development of dietary guidance and advice has been largely neglected, and there are only limited data exploring the impact on research outcomes (*Kipnis et al., 2002*). It seems to be tenable that these limitations are a key contributor to

**Table 1.** Characteristics of dietary bioactives used as model system of dietary compounds to investigate the limitation of using single-point estimates to assess intake and investigate health outcomes in nutrition research.

| Dietary compound | Dietary distribution | Factors for variability | Biomarker of intake | Potential health effect |
|---|---|---|---|---|
| Flavan-3-ols | Tea, apple, and cocoa-derived products | Cultivar, agricultural conditions, storage, and processing | Urinary concentrations of gut microbiome-derived flavan-3-ol metabolites (phenyl-γ-valerolactone metabolites) (*Ottaviani et al., 2018a*) | Reduce cardiovascular events and deaths (*Sesso et al., 2022*. Reduce blood pressure *Ottaviani et al., 2018b*)<br><br>Improve cognitive performance (*Sloan et al., 2021*) |
| (–)-Epicatechin | Tea, apple, and cocoa-derived products | Cultivar, agricultural conditions, storage, and processing (including epimerisation) | Urinary concentrations of structural-related metabolites derived from phase II conjugation (*Ottaviani et al., 2019*) | Improve vascular function (*Schroeter et al., 2006*; *Dicks et al., 2022*) and reduce blood pressure (*Ottaviani et al., 2018b*) |
| Nitrate | Vegetables, drinking water | Depends on a wide range of environmental factors such as fertilisation, light exposure, and water supply | Urinary nitrate status can be used as a surrogate marker of intake (*Green et al., 1981*; *Pannala et al., 2003*; *Smallwood et al., 2017*) | Dietary nitrate can reduce blood pressure (*Larsen et al., 2006*) |

the inconsistent and often contradictory outcomes of nutrition research and dietary guidance, which have received a high level of public attention and significant criticism in recent years (*Ioannidis, 2018*).

The European Prospective Investigation into Cancer and Nutrition (EPIC) Norfolk study (n = 25,618, data available for 18,684; *Day et al., 1999*) is ideally suited to investigate the impact of the variability in bioactive content on nutritional research because it has detailed dietary data based on the combination of self-reporting and food composition data, nutritional biomarkers, as well as health endpoints collected at the same time. Bioactives are food constituents that are not considered essential to human life but can affect health and are therefore extensively investigated (*Ottaviani et al., 2022*). We used three dietary bioactives as model compounds, including flavan-3-ols, (–)-epicatechin, and nitrate (*Table 1*) as (i) there are widely used food composition data tables used to estimate their dietary intake (*Figure 1*); (ii) there exist suitable nutritional biomarkers, which can provide accurate information on actual intake *Kaaks et al., 1997*; and (iii) there are data from dietary intervention studies that support associations between intake and health outcomes (*Larsen et al., 2006*; *Ottaviani et al., 2018b*; *Table 1*). For the purpose of this investigation, we determined bioactive intakes in a single cohort using data and samples collected at the same time. We used two different methods: the commonly deployed approach based on combing self-reported dietary intake with data from food composition tables (DD-FCT) as well as a method based on measuring nutritional biomarkers in urine samples (biomarker method). In the context of the first approach, we also considered taking into consideration nutrient content variability data provided by current food content tables. This was achieved by not only using single-point estimates (mean values) as is common practice, but also by considering reported content ranges (*Blekkenhorst et al., 2017*; *Rothwell et al., 2013*), using a probabilistic-type modelling approach. While our study focuses on bioactives, it is likely that the results will also apply to nutrients and other food constituents with high variability such as minerals, where more than twofold variabilities were previously observed (*Wilkinson and Perring, 1961*), and other nutrients, including macronutrients such as fatty acids (*Reig et al., 2013*; *Schwendel et al., 2015*). The findings of our study aim to test whether current approaches most often relying on the standardised, single-point food content estimates obtained from food composition data can provide useful estimates of actual dietary intake and allow the investigation and meaningful interpretation of associations with health.

## Results
### Impact of bioactive content variability when assessing dietary intake
The intake of an individual nutrient or bioactive is usually calculated by using self-reported dietary data and the mean food content as single-point estimate. While the high variability in food composition is well known and recognised as a source of bias (*National Research Council et al., 1986*), this is rarely

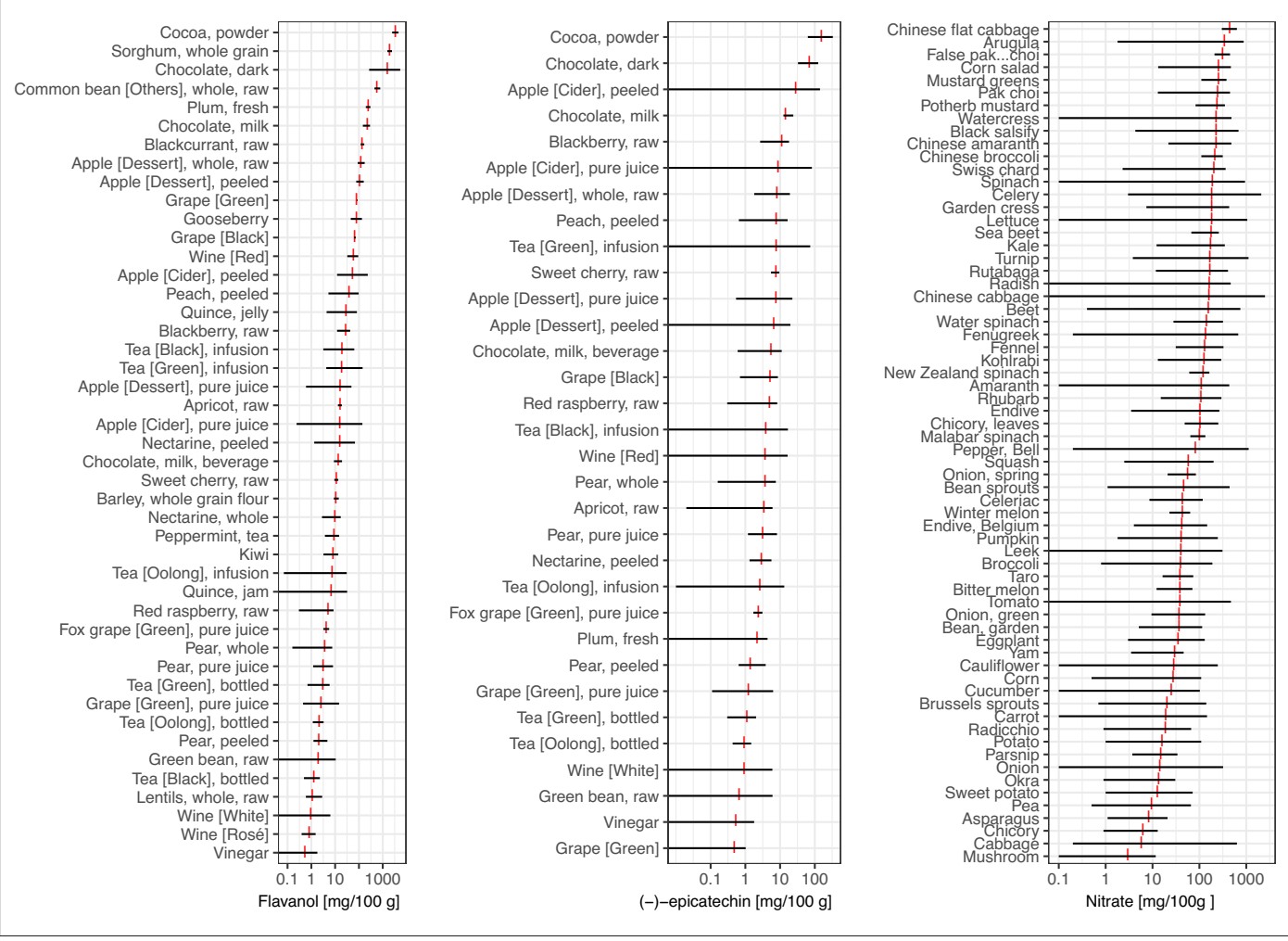

**Figure 1.** Variability in flavan-3-ol, (–)-epicatechin (*Rothwell et al., 2013*), and nitrate (*Blekkenhorst et al., 2017*) content of foods commonly eaten. Data show the range of food content (black) and mean (red).

acknowledged in such estimates and often assumed to have only a little impact due to a regression to the mean. However, there is a paucity of data investigating the actual impact of this variability on estimated intakes. We estimated the potential impact of the variability in flavan-3-ols, (–)-epicatechin, and nitrate food content on estimated intakes of the respective compounds and compound classes in 18,684 participants of EPIC Norfolk for whom all relevant data were available (*Table 2*). *Table 3* shows a comparison of estimated intakes when calculated using the DD-FCT approach with mean food content, as is current practice, as well as minimum and maximum reported food content. These results demonstrate a large uncertainty in estimating actual intake when taking the large variability in bioactive content into consideration. In comparison to the uncertainty introduced by the variability in food composition, the uncertainty associated with the use of self-reported methods of 2–25% (*Stubbs et al., 2014*) appears to be small. There is an overlap in the possible range of bioactive intake between study participants (*Figure 2*), making it difficult to identify low and high consumers or to rank participants by intake (see also below). These results show that bioactive content variability significantly contributes to the uncertainty in the estimation of dietary intake, even more than the error incurred by self-report methods that have attracted a lot of attention and discussion in nutritional research (*Subar et al., 2015*).

## Impact of food composition variability when assessing relative intake

In many studies, relative, instead of absolute, intakes, for example, quintiles, are used (*Altman and Bland, 1994*). It is assumed that the relative intake is less affected by measurement error than absolute

**Table 2.** Study population and baseline characteristics of 18,684 participants of EPIC Norfolk, for whom all data were available.

Data shown are mean (SD) or absolute number and proportion. Data for urinary nitrate was available for 1027 samples.

|  | Women | Men |
| --- | --- | --- |
| n | 10,167 | 8517 |
| Age (years) | 59 (9) | 59 (9) |
| Body mass index (kg/m$^2$) | 26.1 (4.2) | 26.4 (3) |
| Systolic blood pressure (mmHg) | 134 (19) | 138 (17.6) |
| *Physical activity* |  |  |
| Inactive | 2997 (30%) | 2577 (30%) |
| Moderately inactive | 3258 (32%) | 2096 (25%) |
| Moderately active | 2309 (23%) | 1990 (23%) |
| Active | 1603 (16%) | 1854 (22%) |
| *Smoking status* |  |  |
| Current | 1121 (11%) | 998 (12%) |
| Former | 3250 (32%) | 4647 (55%) |
| Never | 5796 (57%) | 2872 (34%) |

intake, and thus can mitigate some of the limitations of estimating dietary intake (*Streppel et al., 2013*). We therefore investigated how the ranking of participants is affected by the variability in bioactive content and compared the relative intake of participants with low (p25 – based on mean bioactive content), medium (p50), and high (p75) intake. Bioactive content variability was introduced in the analysis using an approach similar to probabilistic modelling by sampling randomly from the distribution of possible food composition for each food consumed by each participant. *Figure 3* shows the result of 10,000 of such simulations. They suggest that the high variability in bioactive content makes estimates of relative intakes unreliable. Indeed, depending on the actual food consumed, the self-same diet could put the self-same study participant in the bottom or top quintile of intake. This suggests that it is difficult to obtain reliable relative intakes from dietary data alone, and that ranking by those data is unreliable.

In order to confirm the findings of our simulations, we compared relative intakes estimated using data from DD-FCT and biomarker method. The biomarkers used in this study (*Green et al., 1981*; *Ottaviani et al., 2018a*; *Ottaviani et al., 2019*; *Pannala et al., 2003*; *Smallwood et al., 2017*) have been validated and characterised previously (*Table 1*) and are suitable to estimate relative intake (*Keogh et al., 2013*). Like the 24 hr dietary recall data used here, biomarkers reflect acute intake. The intake estimated from the DD-FCT method was calculated using the common approach based on the mean bioactive content in databases. The association between this self-reported intake and biomarker is weak, with a maximum Kendall's $\tau$ of 0.16 for (–)-epicatechin and lower for flavan-3-ols

**Table 3.** Intake of different bioactive compounds in EPIC Norfolk (median and interquartile range) when determined using different food composition data.

Results are shown for estimates calculated using minimum, mean, and maximum food content and self-reported dietary data based on 24 hr diet recall (24HDR).

| | Bioactive intake (mg/day) | | |
| --- | --- | --- | --- |
| | Minimum food content | Mean food content | Maximum food content |
| Flavan-3-ols | 48 (28–82) | 120 (70–190) | 329 (172–451) |
| (–)-Epicatechin | 1.5 (1.0–2.5) | 19 (9–25) | 33 (65–100) |
| Nitrate | 5.5 (4.6–57) | 100 (80–124) | 204 (151–305) |

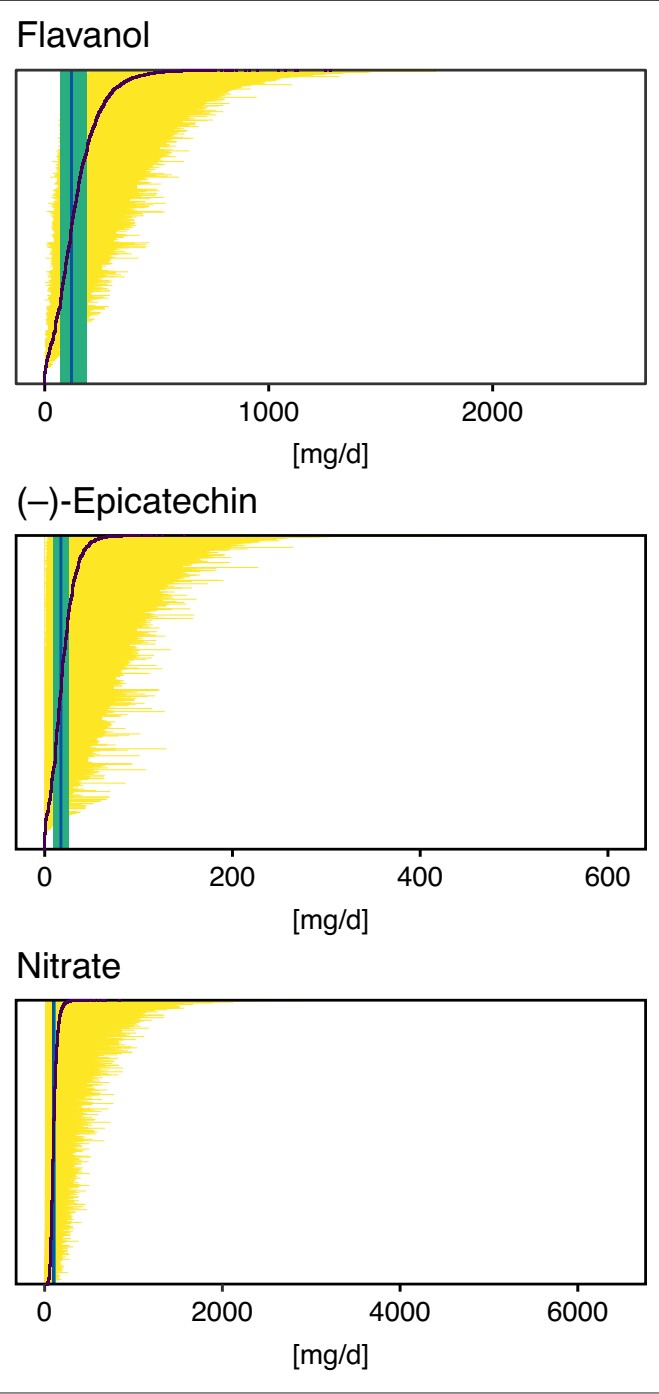

**Figure 2.** Possible intake ranges of flavan-3-ols, (–)-epicatechin, and nitrate in each individual study participant displayed from low to high possible bioactive intake level. Range of bioactive intake was calculated using an approach similar to probabilistic modelling by sampling randomly from the distribution of possible food composition (n = 10,000 iterations). Intake based on mean bioactive content, as is common practice, is indicated by a black line. Green line shows the median intake of the entire cohort and the green box the interquartile range.

(0.06) and nitrate (–0.05). *Figure 4* illustrates this by comparing respective quantiles of intake as these are commonly used to categorise relative intake. The data show very modest agreement between the two measurement methods (only 20–30% of participants assigned to the same quantile) and confirm that ranking is not suitable to address the measurement error and uncertainty introduced by the high

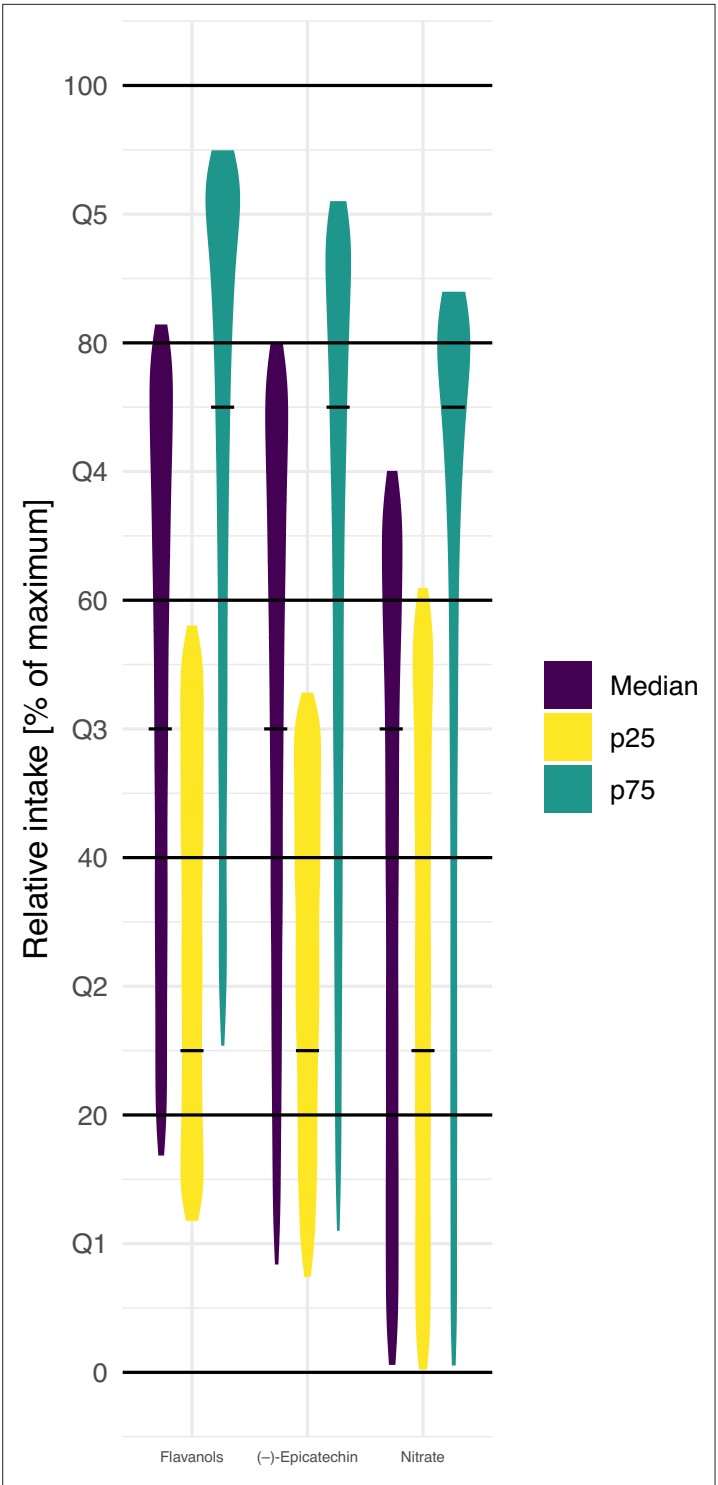

**Figure 3.** Simulation of the effect of variability in food composition on relative intakes of flavanols, (–)-epicatechin, and nitrate of EPIC Norfolk participants with low (25th centile, p25), medium (median), and high (75th centile, p75) estimated intake of bioactive (based on 24 hr diet recall [24HDR] and mean food content – indicated by the black line). Data shown are relative intake (100% is the maximum intake) of 10,000 simulations.

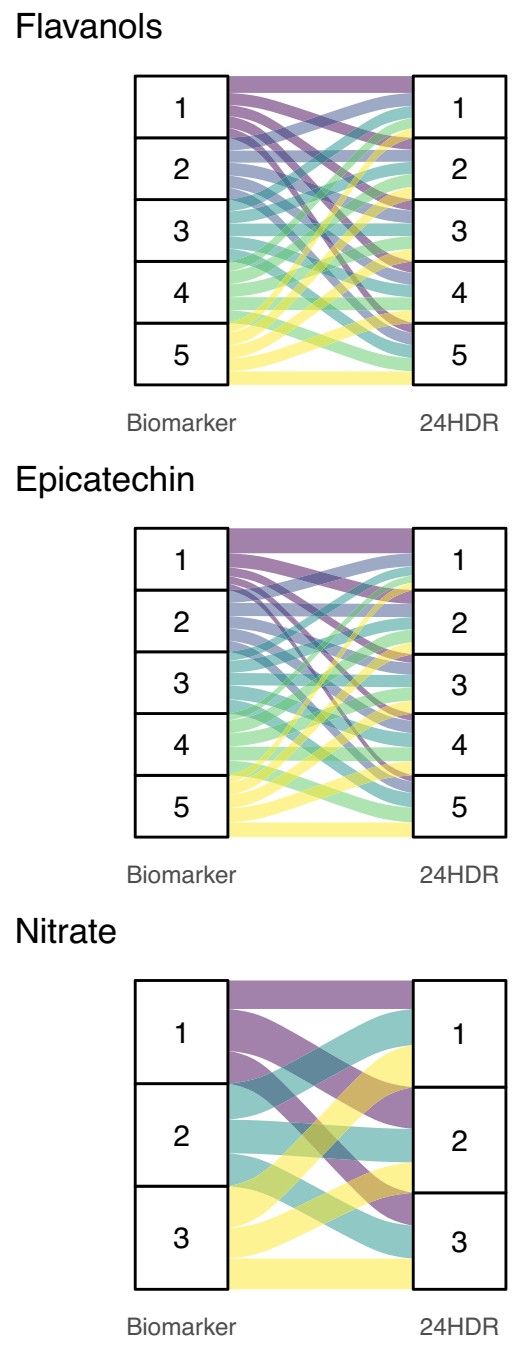

**Figure 4.** Alluvial plots comparing quantiles of bioactive intake estimated with biomarkers (left) and 24 hr dietary recall with food-composition data (24HDR – DD-FCT method, right). Bands of the same colour show participants in the same biomarker-estimated quantile of intake and their respective quantile of intake based on the DD-FCT method. 24HDR estimated quantiles of intake were determined using the common approach of using mean content of flavan-3-ols, (–)-epicatechin, and nitrate in each food item as reported in databases.

variability in bioactive content. Overall, this shows that relying on a single value of bioactive content in food for all participants introduces bias when assessing relative intake of dietary compounds.

## Impact of bioactive content variability on the estimated association between intake and health endpoints

We showed earlier that the high variability in food composition has an impact on the estimates of intake using the DD-FCT method. However, it is not known whether this affects estimated associations between intake and health endpoints. Here, we use simulations to explore how the variability in food compositions affects such estimates in a 'vibration of effects'-type approach (*Patel et al., 2015*) and compare these with the results derived from biomarker-estimated intakes. We use the cross-sectional association with blood pressure as example as all three compounds have a well-established acute effect on vascular function (*Larsen et al., 2006*; *Ottaviani et al., 2018b*; *Schroeter et al., 2006*).

*Figure 5* shows the high variability in estimated associations for all three bioactives under investigation. Each estimate shown is based on identical dietary data and thus represents a possible true association between bioactive intake and blood pressure, depending on the actual bioactive content. It is noticeable that we observe a Janus effect with the DD-FCT method-estimated associations being in opposing positions. This is very noticeable for nitrate, where the estimated differences in blood pressure range from –1.0 (95% CI –1.6 to –0.4) mmHg between the bottom and top decile of intake, suggesting a potentially beneficial, to 0.8 (0.2; 1.4) mmHg, suggesting a potentially detrimental effect on health. As the actual food composition is unknown, it is not possible to obtain a reliable estimate of this association or even identify the likely direction of such an association. Using the mean bioactive content, as is common practice, does not resolve this challenge. Biomarker-derived data, while not deprived of limitations but certainly not affected by the factors that modulate variability in food content in the DD-FCT method, show a strong and significant inverse association between intake and blood pressure, and this association would have been missed when relying exclusively on dietary data.

These results show that the variability in bioactive content can impact the estimated associations between the DD-FCT method intake assessments

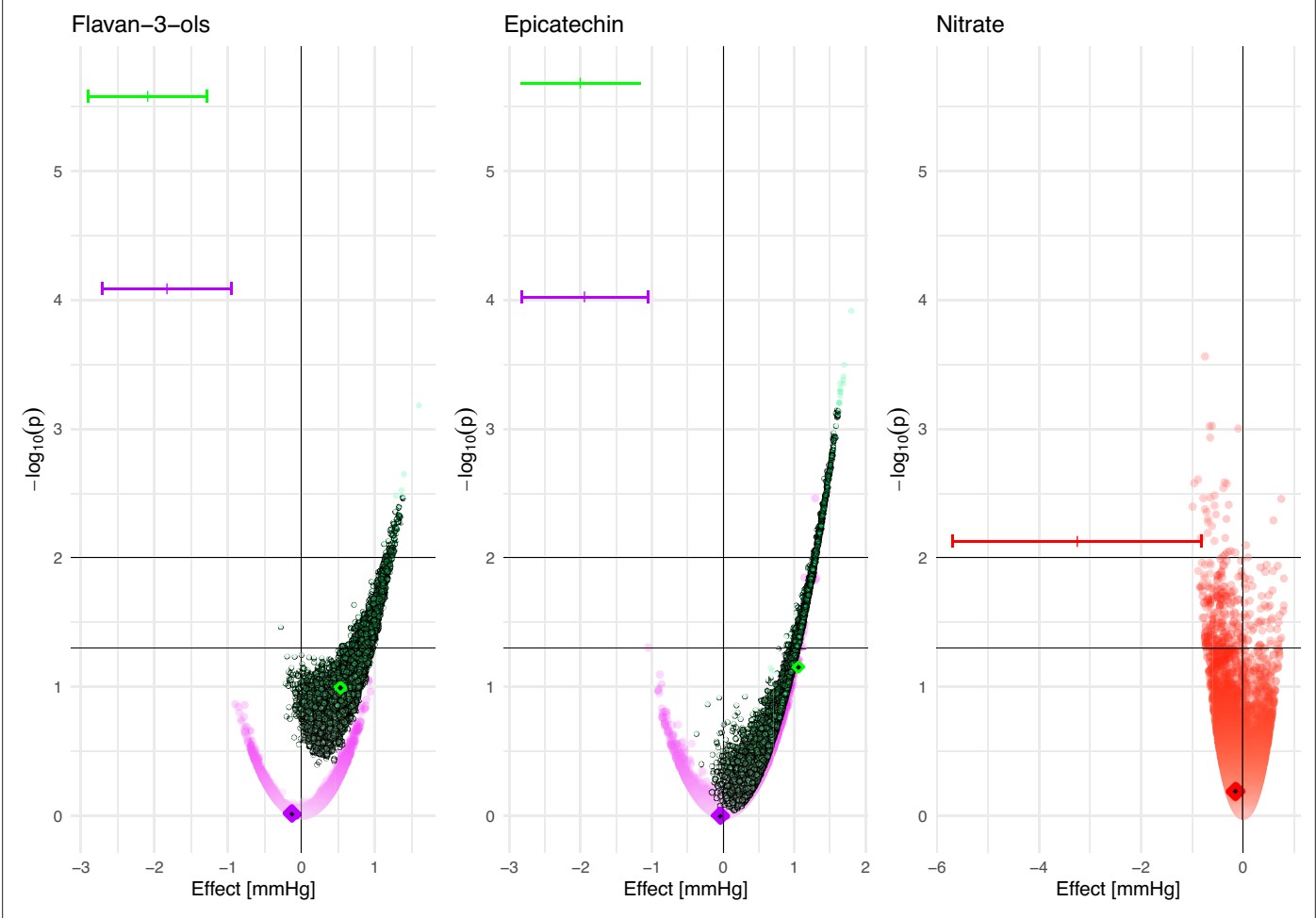

**Figure 5.** Association between estimated bioactive intake (flavan-3-ols, (–)-epicatechin, and nitrate, based on the 24 hr dietary recall and food composition data [DD-FCT method]) and systolic blood pressure at baseline (estimated difference between low [p10] and high [p90] intake and p-value for Wald-test (as -log10(p)) in men [purple], women [green], and all participants [red]). Data are based on 10,000 simulations and adjusted for age, body mass index (BMI), plasma vitamin C, smoking status, physical activity, and self-reported health at baseline; additionally for menopausal status for women and sex for nitrate. Results based on the intake estimated by simulating food content within minimum and maximum food content reported in databases (circle), intake based on mean food content as reported in databases (diamond), and intake based on biomarker data (|). 95% CI is shown for biomarker only.

and health endpoints. It demonstrates that even when using the self-same food intake data, differences in bioactive content can result in diametrically opposite results. Considering that most studies investigating the associations between the intake of bioactives and health do not take variability in food composition into account, it is likely that many reported associations are unreliable.

## Discussion

In this study, we investigated how the variability in food composition affects nutritional research. Our results, based on three bioactives, show that the variability in food composition represents a significant factor that needs to be taken into consideration. The use of single-point estimates of food composition data represents a significant oversimplification that yields unreliable data as the actual intake can be considerably different from the estimated intake. This is often exacerbated by errors that arise from imputing data into food composition tables from analyses conducted in different countries or by changes in the formulation of foods from food manufacturers. These findings are not only important for observational studies, but also for dietary intervention studies, where such methods are often used to estimate background dietary intake and the trial designs of a given intervention.

Our results show that the variability in food composition makes reliable estimates of both absolute and relative intake of bioactives challenging and potentially highly unreliable when solely relying on a combination of self-reported dietary data and food composition databases. This significant limitation is further applied by the bias introduced through limitations of dietary assessment itself (e.g., reporting bias). Thus, any associations between intake and health outcomes derived from using many current approaches are unreliable. Indeed, for the three food constituents under investigation, we found a Janus-like effect with negative and positive associations using the self-same food consumption data and food content within the reported range. This might help explain why nutritional advice given to the general public can feel inconstant, and even contradictory, at times, especially when observing the evolution of advisory statements on the same foods, food groups, or nutrients over time.

Validated nutritional biomarkers, especially recovery biomarkers (*Kaaks et al., 1997*), can provide a reliable estimate of nutrient and bioactive intake as they are based on their systemic presence and do not rely on assumptions about food composition data. In contrast to the duplicate diet method, which relies on the full analysis of all foods consumed, biomarkers provide better information to investigate the associations between intake and health biomarkers as they reflect not just consumption but also nutrient–nutrient interactions and bioavailability, which can affect the systemic presence of many bioactives (*Ottaviani et al., 2023*).

The application of biomarkers to assess nutrient intakes is not without limitations; however, these limitations can be addressed today. Doing so is often of greater technical feasibility and tenably delivers greater overall improvements than to address the limitations of current non-biomarker-based approaches, including self-report bias and the imprecisions and other limitations of today's food composition databases. This is due to the fact that even successfully mitigating limitations related to reporting bias and food composition analyses does not address the inherent shortfalls of non-biomarker-based methods. These include the unknown impact of pre- and postprandial nutrient–nutrient interactions, inter-subject variations in absorption and metabolism, and the often unknown effects of food processing, preparation, and storage on nutrient composition of foods, which can be addressed through the use of biomarkers.

An important challenge when developing biomarker-based methods for assessing intake is related to the inter-subject variance in the absorption and metabolism of a specific nutrient or bioactive. It is therefore important to establish a physiological link as well as a strong statistical association between intake and biomarker, such as has been done for the biomarkers used in this study (*Ottaviani et al., 2018a*; *Ottaviani et al., 2019*; *Pannala et al., 2003*). Biomarkers need to be evaluated using data of actual bioactive intake and should not rely on published food composition data due to the limitations described above. Except for recovery biomarkers in 24 hr urine, most biomarkers are used to provide relative intake data in order to rank participants according to intake. Our results however show that biomarker-based ranking of intake is much more reliable than the rankings based on methods relying on self-reported data and food composition databases.

High variability in food composition has been described for a range of compounds, for example, for the fatty acid composition of dairy (*Moate et al., 2007*; *Stergiadis et al., 2019*) or vitamins (*Phillips et al., 2018*). There is also a longitudinal variation in food composition, in particular due to changes to cultivars, production practices, and distribution and processing methods (*Davis et al., 2004*), and climate change is likely to exaggerate this (*Macdiarmid and Whybrow, 2019*). Thus, bioactive and nutrient content variability must be taken into consideration when choosing the tools to investigate not only dietary bioactives but also micro- and macronutrients.

The methods commonly used to address measurement error in nutritional research, such as regression calibration (*Spiegelman et al., 1997*), are not suitable to address the limitations introduced by the high variabilty. These methods rely on a known relationship between reported and actual intake in a calibration study to predict the actual intake in a larger cohort. However, the composition of the food actually consumed by participants is impossible to predict as it depends on a range of factors, many of which are unknown to consumers and researchers as outlined in the introduction.

There are of course also other sources of bias and variability that affect dietary assessment. We excluded those from our study as much as possible by using the self-same dietary data for all analyses using only acute intake data (24 hr dietary recall and spot urine samples) and an endpoint that is affected directly by intake. This allows us to attribute our findings mainly to the variability in food composition.

In our study, we used the identical dietary data to investigate the impact of the variability in food content. This allowed us to exclude other sources of variability in dietary assessment, in particular misreporting of dietary intake. We also used measures of acute intake (24 hr dietary recalls and spot urine samples) and used a health endpoint that is directly affected by intake.

## Prospective studies

In our study, we focused on cross-sectional associations between bioactives and blood pressure as the acute effect of these compounds is well established. It is expected that the variability in food composition affects prospective analyses more than cross-sectional analyses: in addition to the variability in food content, the composition of foods changes over time (*Davis et al., 2004*; *White and Broadley, 2005*).

## Biomarker-predicted dietary patterns

The high variability in the content of dietary compounds in food has also implications for the development of biomarkers for individual foods or dietary patterns. A number of biomarkers have been proposed to estimate the intake of individual foods, for example, proline-betaine as biomarker of citrus fruit intake (*Gibbons et al., 2017*), but the content in citrus fruits is highly variable (14.3—110 mg/100 mL in various citrus fruit juice; *Lang et al., 2017*), and it is thus not possible to estimate actual food intake without using foods in which the content of the dietary compound to use as a biomarker is standardised.

The same applies to metabolomics-based biomarkers of dietary patterns. They are usually developed under highly standardised conditions and reflect the composition of the foods consumed during these studies. Changes in the composition of these foods affect the concentration of metabolites and thereby reduce the reliability of metabolite-based biomarkers of individual foods or dietary pattern. This diminishes the suitability of such markers for longitudinal or multicentre studies where a high variability in food composition is likely. These limitations do not apply to the development of biomarkers of specific bioactives or other nutrients as the variability of bioactive and nutrient content is reflected in the variation of biomarker levels.

## Effect on dietary recommendations and risk assessment

The findings presented in this work have a considerable impact on dietary recommendations and guidelines. Our data clearly show that the results based on the DD-FCT method are likely to be biased and unreliable. Dietary recommendation based on such data emanating from that approach are therefore also likely to be unreliable and misleading. However, the high variability in food composition also has an impact on the translation of health-based guidance values into food-based dietary recommendations. For example, the amount of flavan-3-ols required to achieve a vasculoprotective effect according to the EFSA health claim is 200 mg/day (*EFSA Panel on Dietetic Products, Nutrition and Allergies, 2014*). When using mean food composition data (*Rothwell et al., 2013*), this could be achieved by five cups of tea. However, when using the lowest reported food content, at least 22 cups of tea would have to be consumed to meet the recommended intake. Similarly, 5–6 apples would be sufficient to consume the 50 mg/day (–)-epicatechin assumed to be sufficient to improve vascular function (*Ellinger et al., 2012*; *Hooper et al., 2012*) when using mean food content, but it could be up to 27 when assuming a low content in food. In this manner, it would not be possible to determine whether or not a population is already meeting dietary recommendation for flavan-3-ols without the development of biomarker-based methods (*Crowe-White et al., 2022*).

These findings also have an impact on the risk assessment of food components, in particular those that are naturally present in foods and used as additives such as nitrates (*Mortensen et al., 2017*) or phosphates (*Younes et al., 2019*). Results from observational studies and intervention studies relying on food content data will be affected by inaccurate assessment of intake as described above. More importantly, however, the exposure assessment will be affected by the variability of data, with consequences for consumers and food producers as an overestimation of exposure could result in unnecessary restrictions in use, whereas an underestimation could put consumers at risk. For example, in EPIC Norfolk, none or only a very few study participants exceed the ADI (acceptable daily intake) of 3.7 mg/kg BW/day (*Mortensen et al., 2017*) for nitrate when estimating intake with minimum and mean food content, respectively. However, when using the maximum food content, one-third of study

participants exceed the ADI for nitrate, and almost 10% exceed it twofold. Each of these scenarios would result in very different actions by risk managers due to the different impact on population health, and in the latter case more stringent restrictions were necessary.

## Conclusions

Our data suggest that the results of many interventional and observational nutrition studies using dietary surveys in combination with food composition data are potentially unreliable and carry greater limitations than commonly appreciated. As these studies are used to derive evidence-based dietary recommendations and disease risk assessments, their limitations could have a considerable impact on public health. We demonstrated that the results relying solely on food composition data not only failed to identify beneficial associations between three bioactives and blood pressure, but even suggested possible adverse associations. It is highly likely that the findings of this nature are not limited to the model compounds that served as examples in our investigation here but broadly apply to other dietary components as well. Given the importance of diet in the maintenance of health and disease risk reduction, it is crucial to address this limitation: both by revisiting previous studies and by taking these limitations into consideration in future studies. We think it is essential to develop and use nutritional biomarkers to determine actual nutrient intakes that ensure more reliable and actionable insights. This means that the development of more and better biomarkers for accurate dietary assessment remains crucial (*Prentice, 2018*). The challenges associated with developing biomarker-based approaches are not insignificant, but the technical capabilities required are broadly available today, and the advantages of deploying improved approaches to establishing the links between diet and health are so significant, timely, and needed that it should become a standard tool in nutrition research.

## Methods

### Study population

Between 1993 and 1997, 30,447 women and men aged between 40 and 79 years were recruited for the Norfolk cohort of the EPIC study, and 25,639 attended a health examination (*Day et al., 1999*). Health and lifestyle characteristics, including data on smoking, social class, and family medical history, were assessed by a questionnaire. Height and weight measurements were collected following a standardised protocol by trained research nurses. Physical activity, representing occupational and leisure activity, was assessed using a validated questionnaire (*Wareham et al., 2002*). Blood pressure was measured using a non-invasive oscillometric blood pressure monitor (Acutorr; Datascope Medical, Huntingdon, UK; validated against sphygmomanometers every 6 months) after the participant had been seated in a comfortable environment for 5 min. The arm was horizontal and supported at the level of the mid-sternum; the mean of two readings was used for analysis. Non-fasting blood samples were taken by venepuncture and stored in serum tubes in liquid nitrogen. Serum levels of total cholesterol were measured on fresh samples with the RA 1000 autoanalyser (Bayer Diagnostics, Basingstoke, UK). Plasma vitamin C was measured using a fluorometric assay as described previously (*Khaw et al., 2001*). Spot urine samples were collected during the health examination and stored at –20°C until analysis. The study was approved by the Norwich Local Research Ethics Committee, all participants gave written, informed consent, and all methods were carried out in accordance with relevant guidelines and regulations.

Diet was assessed by 7-day diary (7DD), whereby the first day of the diary was completed as a 24 hr recall (24HDR) with a trained interviewer and the remainder completed during subsequent days. Diary data were entered, checked, and calculated using the in-house dietary assessment software DINER (Data into Nutrients for Epidemiological Research) and DINERMO (*Welch et al., 2001*). Flavan-3-ol intake (the sum of epicatechin, catechin, epicatechin-3-O-gallate, catechin-3-O-gallate, and proanthocyanidins) was estimated as described previously *Vogiatzoglou et al., 2015*; minimum and maximum estimated flavan-3-ol intake was estimated using the minimum and maximum food content data provided by Phenol Explorer und USDA databases (*Rothwell et al., 2013*). Nitrite and nitrate intake, based on minimum, maximum, and mean food content, were estimated using a database published previously (*Blekkenhorst et al., 2017*).

## Nutritional biomarker

### Flavan-3-ols and (–)-epicatechin

We used two different biomarkers to estimate flavan-3-ol and (–)-epicatechin intake: $gVLM_B$ that includes the metabolites 5-(4'-hydroxyphenyl)-γ-valerolactone-3'-glucuronide (gVL3G) and 5-(4'-hydroxyphenyl)-γ-valerolactone-3'-sulphate (gVL3S), and $SREM_B$ that includes the metabolites (–)-epicatechin-3'-glucuronide (E3G), (–)-epicatechin-3'-sulphate (E3S) and 3'-methoxy-(–)-epicatechin-5-sulphate (3Me5S). $gVLM_B$ are specific for estimating the intake of flavan-3-ols in general, including (±)-epicatechin, (±)-catechin, (±)-epicatechin-3-O-gallate, (±)-catechin-3-O-gallate, and procyanidins and excluding the flavan-3-ols gallocatechin, epigallocatechin, gallocatechin-3-O-gallate, epigallocatechin-3-O-gallate, theaflavins, and thearubigins (*Ottaviani et al., 2018a*). $SREM_B$ are specific for (–)-epicatechin intake (*Ottaviani et al., 2019*). Spot urine samples were collected during the baseline health examination and stored in glass bottles at –20°C until analysis. Stability analyses confirmed that biomarkers are stable under these conditions (*Ottaviani et al., 2019*). Samples were analysed in random order using the method described previously (*Ottaviani et al., 2019*), with automated sample preparation (Hamilton Star robot; Hamilton, Bonaduz, Switzerland). Concentrations below the lower limit of quantification (LLOQ, 0.1 µM) were used for the analysis to avoid the bias of substituting a range of values by a single value. Concentrations were adjusted by specific gravity for dilution as the endpoint of the analysis, systolic blood pressure, was strongly correlated with urinary creatinine. We used specific gravity to adjust for dilution previously when there was a strong association between creatinine and study endpoint (*Bingham et al., 2007*).

Flavan-3-ol and (–)-epicatechin biomarker data, as well as data for all other variables, were available for 18,864 participants. Data for nitrate biomarker were available for 1027 participants.

### Nitrate

Urinary nitrate concentration, adjusted for dilution by specific gravity, was used as a biomarker of nitrate intake, as between 50 and 80% of dietary nitrate are recovered in urine, whereas endogenous production is relatively stable at 0.57 (95% CI 0.27–0.86) mmol/day (*Green et al., 1981*; *Packer et al., 1989*). A random subset of 1027 samples were analysed by ion chromatography with colorimetric detection (NOx Analyser ENO-30, EICOM, San Diego, CA).

## Simulation of variability

We conducted 10,000 simulations to explore the impact of the variability on bioactive content. For each simulation, we assigned each participant a possible intake of total flavan-3-ol, (–)-epicatechin, and nitrate based on their self-reported dietary intake and the minimum and maximum reported content of each compound in the foods consumed. The data available do not suggest that food composition follows a normal distribution, and we therefore assumed a uniform distribution.

## Data analysis

Data analyses were carried out using R 3.6 (*R Development Core Team, 2023*), using the packages rms (*Harrell, 2023*) for regression analyses, ggplot2 (*Wickham, 2016*) and gridExtra (*Auguie, 2017*) for the generation of graphics. Regression analyses were conducted using ols as regression function. We used the Wald statistics calculated by the *rms* anova function to investigate the relationship between dependent and independent variables, and test for linearity. The *tableone* package (*Yoshida and Bartel, 2022*) was used to prepare tables. Unless indicated otherwise, results are shown with 95% CIs.

### Descriptive statistics

Descriptive characteristics of the study population were summarised using mean (standard deviation) for continuous variables and frequency (percentage) for categorical variables.

## Data transformation

Biomarker data were positively skewed (log-normal distribution), and, therefore, log2-transformed data were used for all analyses. Restricted cubic splines (3 knots, outer quantiles 0.1 and 0.9; using the rcs function; *Harrell, 2023*) were used for all continuous variables unless indicated otherwise.

## Cross-sectional analyses

In cross-sectional analyses, stratified by sex, we investigated the associations between biomarker and 24 hr recall estimated flavan-3-ol, (–)-epicatechin, and nitrate intake (biomarkers adjusted by specific gravity adjusted, dietary data by energy, log2-transformed), as independent variable and systolic and diastolic blood pressure (mmHg) using multiple regression analyses. Analyses were adjusted by age (continuous; years), body mass index (BMI) (continuous, kg/m$^2$), plasma vitamin C, smoking status (categorical; never, ever, former), physical activity (categorical; inactive, moderately inactive, moderately active, active), and health at baseline (self-reported diabetes mellitus, myocardial infarction, cerebrovascular accident). Analyses with flavan-3-ol and (–)-epicatechin as independent variable were stratified by sex, and analyses for women additionally adjusted by menopausal status; analyses with nitrate as independent variable were adjusted by sex and menopausal status.

# Acknowledgements

The EPIC-Norfolk study (doi:10.22025/2019.10.105.00004) received funding from the Medical Research Council (MR/N003284/1 MC-UU_12015/1 and MC_UU_00006/1) and Cancer Research UK (C864/A14136). We are grateful to all the participants who have been part of the project and the many members of the study teams at the University of Cambridge who have enabled this research. The preparation of this paper was supported through a writing retreat funded by the Agriculture, Food and Health research Theme at the University of Reading.

# Additional information

### Competing interests

Javier I Ottaviani, Hagen Schroeter: employed by Mars, Inc, a company engaged in flavanol research and flavanol-related commercial activities. Gunter GC Kuhnle: has received an unrestricted research grant from Mars, Inc. The other author declares that no competing interests exist.

### Funding

| Funder | Grant reference number | Author |
|---|---|---|
| Mars | | Gunter GC Kuhnle |

The funders had no role in study design, data collection and interpretation, or the decision to submit the work for publication.

### Author contributions

Javier I Ottaviani, Hagen Schroeter, Conceptualization, Resources, Formal analysis, Investigation, Methodology, Writing – original draft, Writing – review and editing; Virag Sagi-Kiss, Formal analysis, Methodology; Gunter GC Kuhnle, Conceptualization, Resources, Data curation, Software, Formal analysis, Investigation, Visualization, Methodology, Writing – original draft, Project administration, Writing – review and editing

### Author ORCIDs

Javier I Ottaviani ⓘ https://orcid.org/0000-0002-4909-0452
Virag Sagi-Kiss ⓘ http://orcid.org/0000-0003-3959-6596
Hagen Schroeter ⓘ https://orcid.org/0000-0001-6569-382X
Gunter GC Kuhnle ⓘ https://orcid.org/0000-0002-8081-8931

Joint public review: https://doi.org/10.7554/eLife.92941.3.sa1

Author response https://doi.org/10.7554/eLife.92941.3.sa2

## Additional files

### Supplementary files
• MDAR checklist

### Data availability
The data used in this study is from the EPIC Norfolk cohort. EPIC Norfolk aims to make data and samples as widely available as possible whilst safeguarding the privacy of our participants, protecting confidential data and maintaining the reputations of our studies and participants aims. Information on how to request data from EPIC Norfolk can be found here: https://www.epic-norfolk.org.uk/for-researchers/data-sharing/data-requests/. The code can be obtained from https://gitlab.act.reading.ac.uk/xb901875/reliance-on-self-reporting (copy archived at *Kuhnle, 2024*).

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
