## [Editor Report · eLife assessment]

This **important** study, using three bioactive compounds as a model, demonstrates that estimating the intake of food components based on food composition databases and self-reported dietary data is highly unreliable. The authors present **convincing** data showing the differences in the estimated quantile of intake of three bioactive compounds between biomarker and 24-hour dietary recall with food composition database. The work will be of broad interest to the clinical nutrition research community.

---

## [Referee Report · Joint public review]

Identifying dietary biomarkers, in particular, has become a main focus of nutrition research in the drive to develop personalized nutrition.

The aim of this study was to determine the accuracy of using food composition databases to assess the association between dietary intake and health outcomes. The authors found that using food composition data to assess dietary intake of specific bioactives and the impact consumption has on systolic blood pressure provided vastly different outcomes depending on the method used. These findings demonstrate the difficulty in elucidating the relationship between diet and health outcomes and the need for more stringent research in the development of dietary biomarkers.

The primary strength of the study is the use of a large cohort in which dietary data and the measurement of three specific bioactives and blood pressure were collected on the same day. The bioactives selected have been extensively researched for their health effects. Another strength is that the authors controlled for as many variables as possible when running the simulations to get a more accurate account of how the variability in food composition can impact research findings that associate the intake of certain food components with health outcomes.

---

## [Author Response]

The following is the authors’ response to the original reviews.

We would like to thank the editors and reviewers for their encouraging comments. Reviewer 1 raises an important question regarding the translation of biomarker derived data into dietary recommendations, taking the high variability in food composition into consideration. Unfortunately, there is no straightforward answer as the high variability in food composition means that the number of cups of tea for 200mg of flavan-3-ols will depend on the flavanol content of the tea. A probabilistic modelling approach, as we have used to investigate the impact of food content variability on estimated associations with health outcomes, would be a possible solution. This could provide food based recommendations that would meet a defined intake with a certain probability. However, developing and exploring such models is beyond the scope of this manuscript and we have therefore decided not to include this in our response. We have stated in the manuscript that such a method needs to be developed.

We have addressed the typographical errors and the other comments as follows:

• Line 126 - this is the first mention of DR-FCT and as such it needs to be defined. This was a typo and it was corrected throughout the manuscript. The actual abbreviation is DD-FCT and it is defined in line 78.

• Figure 4 - what exactly is this figure trying to convey to the reader? A better explanation about this figure is needed. Figure legend was updated and extent hoping to increase clarity.

• Figure 5 - Why are the graphs presented differently, meaning why are the data for the flavan-3-ols and epicatechin differentiated for men and women and not nitrate. The sample size for nitrate was too small to stratify in the same way as for flavan-3-ols.

• Line 365 - more information is needed, I am assuming the authors are stating ”The tableone package for R ...”. As requested by the reviewer, additional details are now included.

We have also revised the abstract, the conclusion and the discussion of limitations of the biomarker approach to improve readabilty of the manuscript.